# Assessment of regional body composition, physical function and sarcopenia among peruvian women aging with HIV: A cross-sectional study

**Diego M. Cabrera**[1,2,3], **Mijahil P. Cornejo**[4], **Yvett Pinedo**[5], **Patricia J. Garcia**[3], **Evelyn Hsieh**[1,6]*

**1** Department of Internal Medicine, Section of Rheumatology, Allergy and Immunology, Yale School of Medicine, New Haven, Connecticut, United States of America, **2** Department of Epidemiology of Microbial Diseases, Yale School of Public Health, New Haven, Connecticut, United States of America, **3** Department of Epidemiology, STD, and HIV, School of Public Health, Universidad Peruana Cayetano Heredia, Lima, Peru, **4** Department of Rheumatology, Hospital Nacional Arzobispo Loayza, Lima, Peru, **5** Department of Infectious Diseases, Hospital Nacional Arzobispo Loayza, Lima, Peru, **6** Department of Internal Medicine, Section of Rheumatology, VA Connecticut Healthcare System, West Haven, Connecticut, United States of America

* evelyn.hsieh@yale.edu

**Data Availability Statement:** All dataset from this project is available for data transparency purposes

## Abstract

Management of chronic conditions and optimization of overall health has become a primary global health concern in the care of people living with HIV in the era of highly active antiretroviral therapy (ART), particularly in lower-and-middle income countries where infrastructure for chronic disease management may be fragmented. Alterations in body composition can reflect important changes in musculoskeletal health, particularly among populations at risk for developing fat and muscle redistribution syndromes, such as women with HIV on ART. Given the lack of data on this topic in Latin America and the Caribbean, we designed an exploratory study to measure these outcomes in a population of women aging with HIV in Peru. We conducted a cross-sectional study among Peruvian women with and without HIV aged ≥40 years. Dual X-ray absorptiometry was used to measure trunk and limb lean mass (LM) and fat mass (FM). Physical performance was assessed with the Short Physical Performance Battery (SPPB) and physical strength with a dynamometer. Sarcopenia was assessed based upon EWGSOP criteria. We used linear regression to model associations between body composition, sarcopenia and physical performance scores. 104 women with HIV and 212 women without HIV were enrolled (mean age 52.4±8.2 vs. 56.4±8.8 years, p≤0.001). Among women with HIV, mean years since diagnosis was 11.8±6 and all were on ART. Mean SPPB score was 9.9 vs 10.8 (p<0.001) between both groups. Sarcopenia spectrum was found in 25.9% and 23.1%, respectively. In the multivariable regression analysis, trunk FM and older age were negatively correlated with physical performance among women with HIV. Severe sarcopenia was found among a greater proportion of those with HIV (3.8% vs. 0.9%, p = 0.84), however this finding was not statistically significant. Women with HIV had significantly lower SPPB scores compared to women without HIV, and trunk FM and upper limb LM were independent predictors for the SPPB and Grip Strength tests,

on https://osf.io/qtyv9/?view_only=
b2a8a2cf2e73488ba7b6fde07f30e103.

**Funding:** Dr. Diego M. Cabrera served as a Fogarty Global Health Trainee and is supported by the Fogarty International Center at the National Institutes of Health and the National Institute of Arthritis and Musculoskeletal and Skin Diseases under grant number D43TW010540. The PI of this grant was Albert Ko. Dr. Evelyn Hsieh was supported by the Fogarty International Center under grant number K01TW009995. The funders had no role in study design, data collection and analysis, decision to publish, or preparation of the manuscript. Dr. Cabrera received a stipend as part of the Fogarty Global Health Fellowship sponsored by the Fogarty International Center.

**Competing interests:** The authors have declared that no competing interests exist.

respectively. Larger, prospective studies are needed in Latin America & the Caribbean to identify individuals at high risk for sarcopenia and declines in physical function, and to inform prevention guidelines.

## Introduction

As survival rates for people living with HIV (PLWH) continue to improve globally due to improved access to antiretroviral therapy, management of chronic conditions and optimization of overall health has become a primary concern in the care of this population [1], particularly in lower-and-middle income countries where infrastructure for chronic disease management may be fragmented. Studies have shown that onset of aging-related comorbidities appears to occur in PLWH approximately a decade earlier compared with the general population [2]. These comorbidities may lead to a decline in physical function and vulnerability to poor health outcomes or developing other pathologies [3]. In Latin America & the Caribbean, few studies have focused on the impact of aging-related comorbidities among middle-aged and older PLWH. Therefore there remains an important research priority to characterize the epidemiology and outcomes of these co-morbidities, given the sociocultural and health systems related factors unique to this region [4].

The musculoskeletal (MSK) system is significantly affected by HIV and its treatment. Compared with the general population, PLWH present with accelerated bone mineral density loss, increased incidence of low bone mass and heterogeneous disorders of body composition, often associated with metabolic consequences. The reasons for this are multifactorial, including a chronic inflammatory state, antiretroviral toxicity, immune activation and dysregulation of bone turnover [5,6]. Alterations in body composition, such as changes in muscle mass and fat redistribution, can reflect important changes in bone and MSK health [7]. Changes in muscle mass, together with declines in muscle performance/function and strength, which can be seen with aging, contribute to development of sarcopenia, which has been independently associated with morbidity and mortality [8].

As PLWH age with these comorbidities, the sociodemographic and clinical heterogeneity of older individuals with HIV highlights the relevance of identifying those who are at greatest risk of premature aging [9]. The Short Physical Performance Battery (SPPB) is an objective physical functioning assessment that measures physical performance in older populations and has been extensively used in research settings to identify patients at risk of adverse events [10,11]. Physical function measures have multiple applications among aging adults including identification of those at increased likelihood of poor outcomes, which may allow clinicians to identity high-risk older individuals and prioritize them for preventive interventions [12].

Although more than half of PLWH worldwide currently are women, there is limited data about the specific characteristics of women aging with HIV [13]. Furthermore, no studies have focused on the relationship between body composition changes, sarcopenia and physical function among women with HIV in Latin America and the Caribbean. We aimed to measure physical function, physical strength and sarcopenia among middle-aged and older Peruvian women with and without HIV using well-established rigorous measures, and to explore the association of the outcomes in the two groups with key clinical and sociodemographic risk factors. We hypothesized that in addition to age, certain HIV-associated factors (e.g., CD4 + count, duration of HIV, exposure NRTIs or PIs) would be associated with outcomes among women with HIV in our study.

## Methods

### Study design and participants

We conducted a hospital-based cross-sectional study. Between October 2019 and March 2020, participants presenting for care at a large HIV-clinic from a public hospital in Lima, Peru were consecutively invited to participate in a study regarding musculoskeletal health (osteoporosis, vertebral fractures, regional body composition, physical performance and strength, and sarcopenia) and health-related quality of life. We enrolled women with HIV aged ≥40 years who were on ART and presented for routine HIV-care visits during the study period, and concurrently recruited a control group of HIV-uninfected women aged ≥40 years. We used two strategies to recruit women in the control group, including inviting community-dwelling women from surrounding neighborhoods and as well as advertising the study among hospital staff (administrators, nurses and technicians). Exclusion criteria included pregnancy (assessed with urine pregnancy test at recruitment), previous diagnosis of or treatment for osteoporosis, and insufficient literacy to complete the questionnaire. Written informed consent was obtained in accordance with the procedures approved by the ethics committees from Yale School of Medicine, Universidad Peruana Cayetano Heredia and Hospital Nacional Arzobispo Loayza. The current analyses focus on the muscle-related outcomes of regional body composition, physical performance and strength, and sarcopenia.

### Data collection and measures

Participants completed a self-administered questionnaire including socio-demographic and clinical data. Questionnaires were piloted before enrollment among 10 patients and 10 clinic staff to assure ease of use and clarity. HIV-related characteristics such as $CD4^+$ T-cell count, viral load and ART were obtained from medical records. HIV viral load suppression was defined as <50 copies/ml. Anthropometric measures were obtained using the same standardized stadiometer and weight scale for all participants. Waist and hip circumferences were measured to the closest 0.1 cm using a non-elastic tape measure.

### Body composition measurements

DXA was performed using a Hologic Discovery–WI 2009 series 84453 machine (Hologic Inc, Waltham, MA, USA). Scans of the whole body were performed using the automatic scan mode with participants wearing light clothing according to standard procedures. Body composition measures included trunk and limb lean mass (LM), fat mass (FM) and bone mineral content (BMC), all measured in grams. The fat mass index (FMI), skeletal muscle mass index (SMI) and appendicular skeletal muscle mass index (ASMI) were calculated as total body FM, total body LM and limb LM divided by the square of height (kg/m$^2$), respectively [14]. Daily calibration and long-term DXA stability monitoring were performed according to standard protocols using manufacturer phantoms.

### Tests of physical function and strength

The SPPB was selected as an evidence-based, quick-to-perform physical function assessment. This tool has been used for more than 20 years to assess performance and functional status in aging populations [11]. The SPPB includes timed measures of standing balance in 3 positions (side-by-side, semi-tandem, and tandem), 4-meter gait speed, and time required to stand up and sit down 5 times. The test was performed according to the National Institute on Aging protocol [15]. Each measure was designated a score from 0 to 4, with 0 indicating inability to

complete the test. Composite measures yielded a total score from 0 to 12, where a score ≤8 was used to indicate disability/low physical function [12,16].

Physical strength was assessed with hand grip strength using a JAMAR hydraulic hand dynamometer (Patterson Medical, IL, USA), which is the most widely used dynamometer in the research setting, and has been validated in several different patient populations and countries for measuring grip strength [17,18]. Force was measured in kilograms according to standardized procedures. Participants were asked to sit with their shoulder adducted and neutrally rotated, elbow flexed at 90˚ and with the forearm in a neutral position maximally squeeze the handle of the dynamometer with their dominant hand for three seconds. We set the dynamometer on the second handle position and performed the test three successive times, the average of the three trials was recorded.

## Sarcopenia spectrum

Sarcopenia was defined and staged according to the European Sarcopenia Working Group (EWGSOP) guidelines [19], which recommends the presence of low muscle mass (ASMI≤5.67 kg/m$^2$) and low muscle function (strength [grip strength≤20kg] or performance [SPPB≤8]) for diagnosis of this syndrome [20–22]. The "presarcopenia" stage was defined as low muscle mass without impact on muscle strength or performance. The "sarcopenia" stage was defined as low muscle mass, plus low muscle strength or physical performance and "severe sarcopenia" was defined when all three criteria of the definition were met.

## Statistical analysis

We described the sample characteristics and outcomes using standard frequency analysis, means, standard deviations, and proportions for all variables, as appropriate. Differences between the two groups were examined using independent t-test, Mann-Whitney U test, $x^2$, and Fisher exact test, as appropriate for normally-distributed or non-parametric continuous variables, or categorical variables, respectively.

Unadjusted and adjusted linear regression models were adopted separately for the women with and without HIV to further analyze associations of independent variables (sociodemographic, clinical characteristics and body composition measurements) and physical function (SPPB and grip strength). Two multivariable models were constructed for the independent variables and SPPB, and grip strength separately. We first assessed for normal distribution and possible multicollinearity of the independent variables. Variables with a high correlation coefficient (r>0.40) were deemed collinear and excluded from the models. We fit the multivariate models using backward regression [23], starting with all the variables that showed a hypothesized relationship (p<0.10) in the bivariate model. We then removed non-significant (p>0.05) variables one at a time beginning with the least significant (largest p-value), in order to achieve the most parsimonious model.

Associations between the sarcopenia spectrum (presarcopenia, sarcopenia and severe sarcopenia) and sociodemographic and clinical related factors were explored with logistic regression analysis, following the same steps described above. As body composition measures (and the SPPB and grip strength) are part of the EWGSOP algorithm for identifying sarcopenia spectrum, these were not included in the multivariable regression models for sarcopenia spectrum.

Results were reported as unadjusted and adjusted standardized beta coefficients or odds ratio (OR) with 95% confidence intervals, as appropriate. All statistical analyses were performed using STATA version 16 (StataCorp, College Station, Texas, USA).

## Results

### Sociodemographic and clinical characteristics

A total of 316 women were recruited in this study, including 104 HIV-infected and 212 uninfected women. The mean age of uninfected women was slightly older than women with HIV (56.4±8.8 vs 52.4±8.2 years, p<001). Women with HIV had a lower mean body mass index (BMI) (24.4 kg/m$^2$ vs 27.6, p = 0.05) compared to uninfected women, and a lower percentage of them had high school education or above (65.4% vs 85%, p<0.001) compared to the control group. Postmenopausal women represented 81% of the entire sample (Table 1).

Among women with HIV, the mean time since HIV diagnosis was 11.8±6 years and all women were currently receiving ART (mean duration of treatment = 9.9±5.3 years). Almost all patients were receiving a nucleoside/nucleotide reverse transcriptase inhibitor [including tenofovir disoproxil fumarate (TDF), emtricitabine (FTC), abacavir (ABC), zidovudine (AZT), lamivudine (3TC)]. Just over two thirds were treated with non-nucleoside reverse transcriptase inhibitors [efavirenz (EFV) or nevirapine (NVP)], and just under a third were treated with protease inhibitors [specifically, lopinavir/ritonavir (LPV/r)]. Very few patients (3/104) were treated with an integrase inhibitor [specifically, dolutegravir (DTG)]. The current mean CD4$^+$ T-cell count was 593.2±297.5 cells/mm$^3$ and the mean CD4$^+$ T-cell count nadir was 264.3 ±174.3 cells/mm$^3$. 78.8% had an undetectable viral load.

### Body composition, physical function and strength, and sarcopenia

In terms of body composition measures, women with HIV had decreased total FM (p<0.001) compared to uninfected women. Significant differences were found between FMI (p<0.001), upper limb FM (p = 0.02), lower limb FM (p<0.001) and trunk FM (p = 0.004) between both groups. LM was found to be lower in the upper limb and total body of Women with HIV compared to the control group, but these comparisons were not statistically different.

The overall SPPB score was slightly lower in Women with HIV compared to the control group (10 vs 11, p<0.001). A notable difference was observed in the proportion of patients with SPPB score ≤8, (17% vs 10%, p = 0.001) (Table 2). Small but statistically significant differences were found in SPPB sub-scores between the two groups, including the balance test scores (4 vs 4, p<0.001), gait speed test scores (4 vs 4, p = 0.01) and chair stand test scores (3 vs 3, p<0.001). Average repeated grip strength was similar in the two groups (19.9±5.9 kg vs 19.8 ±5.4 kg).

In terms of the sarcopenia spectrum, 25.9% of HIV-infected and 23.1% uninfected women met criteria for either presarcopenia, sarcopenia or severe sarcopenia, but this difference did not reach statistical significance. Sarcopenia stage was the most common overall (15.4% in HIV-infected vs 16.5% uninfected women), and severe sarcopenia was higher in the HIV-infected subset (3.8% vs 0.9%), but was not statistically different. Moreover, although ASMI was higher among uninfected women compared with women with HIV for both sarcopenia and severe sarcopenia stages, the differences were not statically significant (**Table 2**).

### Associations between clinical characteristics, body composition, and physical function assessments

In terms of physical function, the multivariable analysis among uninfected women showed that higher SPPB score was independently associated with current alcohol use (p = 0.01) and higher lower limb BMC (p = 0.03), and lower SPPB score was significantly associated with older age (p<0.001) and higher BMI (p<0.001) (Table 3). However, in the multivariable

**Table 1. Sociodemographic and clinical characteristics by HIV status.**

| | | HIV-infected | | Uninfected | |
| --- | --- | --- | --- | --- | --- |
| | | N = 104 | | N = 212 | |
| **Sociodemographic characteristics** | | | | | |
| Age, mean *(SD)* | | **52.38** | **(8.15)** | 56.37 | (8.8)¥ |
| Marital status, n(%) | | | | | |
| | Single/divorced/separated/widowed, n(%) | **74** | **(71.15)** | 95 | (44.81)¥ |
| | Married/cohabitant, n(%) | 30 | (28.85) | 117 | (55.19) |
| Ethnicity | | | | | |
| | Caucasian, n(%) | 3 | (2.88) | 24 | (11.32) |
| | Mestizo, n(%) | **97** | **(93.27)** | 185 | (87.26)* |
| | Black, n(%) | 4 | (3.85) | 3 | (1.42) |
| Education level | | | | | |
| | High school education or above, n(%) | **68** | **(65.38)** | 181 | (85.38)¥ |
| **Clinical characteristics** | | | | | |
| Body mass index (kg/m2), mean (SD) | | **26.44** | **(5.1)** | 27.63 | (4.12)* |
| Smoking ever, n (%) | | 28 | (26.92) | 39 | (18.4) |
| Current alcohol use, n(%) | | 25 | (24.04) | 60 | (28.3) |
| Current use of bone nutritional supplements, n(%) | | 5 | (4.81) | 11 | (5.19) |
| Rheumatoid arthritis diagnosis, n(%) | | - | | 6 | (2.83) |
| Postmenopausal status, n(%) | | 80 | (76.92) | 177 | (83.49) |
| **HIV-related clinical data** | | | | | |
| Years since HIV diagnosis median (IQR) | | 12.5 | (7–15) | - | |
| Nadir CD4+ cell count median (IQR) | | 242 | (120.8–400.3) | - | |
| Nadir HIV viral load median (IQR) | | 19895 | (399–134099) | - | |
| Current CD4+ cell count median (IQR) | | 591.5 | (385.8–765.3) | - | |
| Current HIV viral load median (IQR) | | 0 | (0–40) | - | |
| AIDS diagnosis n(%) | | 8 | (7.69) | - | |
| Duration ART exposure (years) median (IQR) | | 10 | (6–14) | - | |
| PI exposure n(%) | | 32 | (30.77) | - | |
| NRTI exposure n(%) | | 103 | (99.04) | - | |
| NNRTI exposure n(%) | | 71 | (68.3) | - | |
| INSTI exposure n(%) | | 3 | (2.9) | - | |
| **Body composition measurements** | | | | | |
| Upper limb | | | | | |
| | FM (kg), mean (SD) | **3.34** | **(1.76)** | 3.79 | (1.62)* |
| | LM (kg), mean (SD) | 3.5 | (0.78) | 3.53 | (0.7) |
| | BMC (kg), mean (SD) | **0.21** | **(0.04)** | 0.22 | (0.04)* |
| Lower limb | | | | | |
| | FM (kg), mean (SD) | **6.53** | **(2.43)** | 8.14 | (2.17)¥ |
| | LM (kg), mean (SD) | 11.47 | (2.18) | 11.47 | (1.72) |
| | BMC (kg), mean (SD) | **0.56** | **(0.12)** | 0.59 | (0.11)* |
| Trunk FM (kg), mean (SD) | | **11.54** | **(3.6)** | 12.71 | (3.22)‡ |
| Total body | | | | | |
| | FM (kg), mean (SD) | **22.66** | **(7.24)** | 25.86 | (6.51)¥ |
| | LM (kg), mean (SD) | 37.33 | (6.33) | 37.65 | (4.88) |
| | BMC (kg), mean (SD) | 1.74 | (0.32) | 1.79 | (0.28) |
| ASMI | | 6.31 | (1.04) | 6.3 | (0.88) |
| FMI | | **9.59** | **(3.02)** | 10.88 | (2.67)¥ |

*(Continued)*

**Table 1.** (Continued)

| | HIV-infected | | Uninfected | |
|---|---|---|---|---|
| | N = 104 | | N = 212 | |
| SMI | 15.76 | (2.42) | 15.84 | (1.89) |

SD, standard deviation; FM, fat mass; LM, lean mass; BMC, bone mineral content; ASMI, appendicular skeletal muscle index; FMI, fat mass index; SMI, skeletal mass index; PI, protein inhibitors; NRTI, nucleoside/nucleotide reverse transcriptase inhibitors; NNRTI, non-nucleoside reverse transcriptase inhibitor; INSTI, integrate inhibitors; CD4+ cell count = cell/mm3; HIV viral load = copies (ml)

*$p \leq 0.05$

‡$p \leq 0.01$

¥$p \leq 0.001$.

model for women with HIV, only trunk FM (p = 0.038) and older age (p = 0.002) were negatively correlated with SPPB score (Table 4).

Regarding physical strength, in the uninfected women multivariable model, greater grip strength was positively correlated with upper limb LM (p<0.001), and negatively correlated with older age (p<0.001) (Table 3). Similarly, among women with HIV, the multivariable

**Table 2. Physical function and strength test and Sarcopenia stages according to HIV status.**

| Characteristics | | Women with HIV | Women without HIV | P-value |
|---|---|---|---|---|
| | | N = 104 | N = 212 | |
| **Physical function and strength tests** | | | | |
| **SPPB** | | | | |
| Overall score, median (IQR) | | 10 (9–11) | 11 (10–12) | **<0.001** |
| | Score ≤8 N(%) | 17 (16.35) | 10 (4.72) | **0.001** |
| Balance test, median (IQR) | | 4 (3–4) | 4 (4–4) | **<0.001** |
| Gait test, median (IQR) | | 4 (3–4) | 4 (3–4) | **0.01** |
| Chair stand test, median (IQR) | | 3 (2–3) | 3 (3–4) | **<0.001** |
| **Grip Strength** | | | | |
| Score (kg), mean (SD) | | 19.93 (5.91) | 19.78 (5.44) | 0.825 |
| **Stages and components of sarcopenia** | | | | |
| **Presarcopenia, n (%)** | | 7 (6.7) | 12 (5.7) | 0.93 |
| | ASMI (kg/m2), mean (SD) | 5.32 (0.3) | 5.24 (0.4) | 0.709 |
| | Grip strength (kg), mean (SD) | 23.57 (1.9) | 25.5 (3.2) | 0.168 |
| | SPPB, mean (SD) | 10.71 (0.8) | 11.66 (0.7) | **0.009** |
| **Sarcopenia, n (%)** | | 16 (15.4) | 35 (16.5) | 0.921 |
| | ASMI (kg/m2), mean (SD) | 5.18 (0.4) | 5.21 (0.3) | 0.793 |
| | Grip strength (kg), mean (SD) | 15.56 (3.5) | 16.46 (3.9) | 0.436 |
| | SPPB, mean (SD) | 10.13 (0.9) | 10.89 (0.9) | **0.008** |
| **Severe sarcopenia, n (%)** | | 4 (3.8) | 2 (0.9) | 0.84 |
| | ASMI (kg/m2), mean (SD) | 4.59 (0.7) | 5.06 (0.1) | 0.399 |
| | Grip strength (kg), mean (SD) | 16.75 (2.5) | 13 (4.2) | 0.226 |
| | SPPB, mean (SD) | 7.5 (1) | 7.5 (0.7) | 1 |

SPPB, short physical performance battery; SD, standard deviation; ASMI, appendicular skeletal mass index.

Presarcopenia: ASMI ≤5.67 kg/m2 and Grip Strength >20kg and SPPB overall score >8.

Sarcopenia: ASMI ≤5.67 kg/m2 and (Grip Strength ≤20kg or SPPB overall score ≤8).

Severe sarcopenia: ASMI ≤5.67 kg/m2 and Grip Strength ≤20kg and SPPB overall score ≤8.

**Table 3. Unadjusted and adjusted linear regression for SPPB and grip strength, women without HIV (N = 212).**

| Independent variables | SPPB | | | | Grip strength test | | | |
|---|---|---|---|---|---|---|---|---|
| | Unadjusted | | Adjusted | | Unadjusted | | Adjusted | |
| | B | 95% C.I. | B | 95% C.I. | B | 95% C.I. | B | 95% C.I. |
| **Body composition measurements** | | | | | | | | |
| Trunk FM | -0.064 | -0.117 to -0.012* | - | - | 0.147 | -0.082 to 0.375 | - | - |
| Trunk BMC | 1.892 | 0.001 to 3.784* | - | - | 15.208 | 7.281 to 23.135¥ | - | - |
| Trunk LM | 0.002 | -0.064 to 0.067 | - | - | 0.386 | 0.110 to 0.662‡ | - | - |
| Trunk %fat | -0.056 | -0.093 to -0.019‡ | - | - | -0.046 | -0.208 to 0.116 | - | - |
| Lower limb FM | -0.056 | -0.135 to 0.023 | - | - | 0.274 | -0.065 to 0.612 | - | - |
| Lower limb BMC | 1.998 | 0.427 to 3.569* | 1.551 | 0.165 to 2.938* | 15.495 | 8.985 to 22.004¥ | - | - |
| Lower limb LM | 0.024 | -0.076 to 0.124 | - | - | 0.818 | 0.403 to 1.234¥ | - | - |
| Lower limb %fat | -0.023 | -0.056 to 0.010 | - | - | -0.041 | -0.183 to 0.100 | - | - |
| Upper limb FM | -0.151 | -0.255 to -0.046‡ | - | - | 0.026 | -0.431 to 0.482 | - | - |
| Upper limb BMC | 3.730 | -0.188 to 7.648 | - | - | 41.255 | 25.254 to 57.256¥ | - | - |
| Upper limb LM | 0.024 | -0.076 to 0.124 | - | - | 1.746 | 0.713 to 2.779¥ | 1.804 | 0.997 to 2.611¥ |
| Upper limb %fat | -0.029 | -0.052 to -0.005* | - | - | -0.097 | -0.197 to 0.003 | - | - |
| Total body FM | -0.032 | -0.058 to -0.006* | - | - | 0.072 | -0.041 to 0.186 | - | - |
| Total body BMC | 0.831 | 0.234 to 1.427‡ | - | - | 5.776 | 3.290 to 8.263¥ | - | - |
| Total body LM | 0.002 | -0.033 to 0.038 | - | - | 0.272 | 0.125 to 0.419¥ | - | - |
| Total body %fat | -0.055 | -0.094 to -0.017‡ | - | - | -0.072 | -0.242 to 0.097 | - | - |
| **Sociodemographic and clinical characteristics** | | | | | | | | |
| Age | -0.067 | -0.084 to -0.050¥ | -0.043 | -0.060 to -0.026¥ | -0.194 | -0.273 to -0.114¥ | -0.161 | -0.227 to -0.094¥ |
| BMI | -0.061 | -0.102 to -0.020‡ | -0.059 | -0.092 to -0.026¥ | 0.105 | -0.074 to 0.284 | - | - |
| Smoking ever | 0.378 | -0.063 to 0.819 | - | - | 0.578 | -1.326 to 2.482 | - | - |
| Current alcohol use | 0.488 | 0.112 to 0.864* | 0.406 | 0.090 to 0.723* | -0.064 | -1.703 to 1.575 | - | - |
| Rheumatoid arthritis diagnosis | -1.767 | -2.776 to -0.758¥ | - | - | -3.895 | -8.316 to 0.526 | - | - |
| Menopause status | -0.750 | -1.202 to -0.298¥ | - | - | -3.338 | -5.274 to -1.402¥ | - | - |
| Bone nutritional supplements, ever | -0.085 | -0.861 to 0.690 | - | - | -0.827 | -4.154 to 2.500 | - | - |

SPPB, short physical performance battery; FM, fat mass; LM, lean mass; BMC, bone mineral content; BMI, body mass index; %fat, percentage of FM

*p ≤ 0.05

‡p ≤ 0.01

¥p ≤ 0.001.

model showed upper limb LM (p<0.001) and higher CD4[+] T-cell count (p = 0.012) were positively correlated with grip strength, and older age (p = 0.13) and total body fat percentage (p<0.001) were negatively correlated with this outcome (Table 4).

## Association between clinical characteristics and sarcopenia

Significant associations were found among the HIV-infected univariate model in terms of marital status, where sarcopenia risk reduced with being married/cohabitant (OR 0.14, p = 0.01). Higher BMI (OR 0.57, p<0.001) and higher current CD4[+] T-cell count (OR 0.99, p<0.5) were also significantly associated with reduced sarcopenia risk, and history of AIDS (OR 10.71, p<0.01) and protease inhibitor (PI) exposure (OR 3.53, p<0.01) were associated with increased sarcopenia risk. Likewise, in the multivariable model, higher BMI (OR 0.56, p<0.001) and being married/cohabitant (OR 0.10, p<0.5) were associated with reduced sarcopenia risk, along with PI exposure (OR 0.19, p<0.01) (Table 5).

**Table 4. Unadjusted and adjusted linear regression for SPPB and grip strength, women with HIV (N = 104).**

| Independent variables | SPPB | | | | Grip strength test | | | |
|---|---|---|---|---|---|---|---|---|
| | Unadjusted | | Adjusted | | Unadjusted | | Adjusted | |
| | B | 95% C.I. | B | 95% C.I. | B | 95% C.I. | B | 95% C.I. |
| **Body composition measurements** | | | | | | | | |
| Trunk FM | **-0.075** | **-0.150 to 0.001*** | -0.076 | -0.148 to -0.004* | 0.082 | -0.240 to 0.404 | - | - |
| Trunk BMC | -1.348 | -4.253 to 1.556 | - | - | **20.482** | **8.871 to 32.093‡** | - | - |
| Trunk LM | -0.061 | -0.142 to 0.021 | - | - | **0.463** | **0.126 to 0.799‡** | - | - |
| Trunk %fat | -0.029 | -0.079 to 0.021 | - | - | -0.157 | -0.365 to 0.051 | - | - |
| Lower limb FM | -0.095 | -0.207 to 0.017 | - | - | 0.012 | -0.466 to 0.490 | - | - |
| Lower limb BMC | 1.038 | -1.279 to 3.55 | - | - | **19.136** | **10.087 to 28.186¥** | - | - |
| Lower limb LM | -0.028 | -0.155 to 0.099 | - | - | **0.924** | **0.422 to 1.427¥** | - | - |
| Lower limb %fat | -0.023 | -0.063 to 0.018 | - | - | -0.165 | -0.333 to 0.004 | - | - |
| Upper limb FM | -0.130 | -0.284 to 0.025 | - | - | 0.325 | -0.332 to 0.983 | - | - |
| Upper limb BMC | 0.667 | -5.810 to 7.143 | - | - | **50.778** | **26.409 to 75.147¥** | - | - |
| Upper limb LM | -0.032 | -0.387 to 0.324 | - | - | **2.521** | **1.107 to 3.936¥** | **2.512** | **1.061 to 3.964¥** |
| Upper limb %fat | -0.018 | -0.051 to 0.015 | - | - | -0.072 | -0.212 to 0.069 | - | - |
| Total body FM | -0.037 | -0.074 to 0.001 | - | - | 0.046 | -0.114 to 0.207 | - | - |
| Total body BMC | 0.294 | -0.559 to 1.148 | - | - | **7.402** | **4.106 to 10.699¥** | - | - |
| Total body LM | -0.018 | -0.061 to 0.026 | - | - | **0.301** | **0.127 to 0.474¥** | - | - |
| Total body %fat | -0.043 | -0.098 to 0.011 | - | - | -0.208 | -0.437 to 0.021 | **-0.407** | **-0.622 to -0.193¥** |
| **Sociodemographic and clinical characteristics** | | | | | | | | |
| Age | **-0.051** | **-0.083 to -0.019‡** | -0.051 | -0.083 to -0.019‡ | **-0.165** | **-0.304 to -0.027‡** | **-0.175** | **-0.305 to -0.450*** |
| BMI | -0.038 | -0.091 to 0.016 | - | - | 0.079 | -0.148 to 0.307 | - | - |
| Smoking ever | 0.453 | -0.159 to 1.065 | - | - | 0.288 | -2.318 to 2.893 | - | - |
| Current alcohol use | **0.645** | **0.015 to 1.274*** | - | - | **3.195** | **0.564 to 5.827*** | - | - |
| Years since HIV diagnosis | 0.022 | -0.022 to 0.067 | - | - | 0.046 | -0.142 to 0.234 | - | - |
| Years since ART start | 0.035 | -0.017 to 0.87 | - | - | 0.091 | -0.128 to 0.311 | - | - |
| Nadir CD4+ count | 0.000 | -0.002 to 0.001 | - | - | 0.005 | -0.001 to 0.012 | - | - |
| Current CD4+ count | 0.000 | -0.001 to 0.001 | - | - | **0.004** | **0.001 to 0.008*** | **0.004** | **0.001 to 0.008*** |
| AIDs diagnosis | -0.688 | -1.708 to 0.333 | - | - | 2.094 | -2,225 to 6.412 | - | - |
| Viral suppression status | 0.473 | -0.192 to 1.139 | - | - | -1.298 | -4.117 to 1.521 | - | - |
| Menopause status | **-0.717** | **-1.353 to -0.81*** | - | - | -0.683 | -3.424 to 2.057 | - | - |
| Bone nutritional supplements, ever | -0.200 | -1.482 to 1.082 | - | - | -2.030 | -7.419 to 3.358 | - | - |

SPPB, short physical performance battery; FM, fat mass; LM, lean mass; BMC, bone mineral content; BMI, body mass index; %fat, percentage of FM.

ART, antiretroviral treatment

*p ≤ 0.05

‡p ≤ 0.01

¥p ≤ 0.001.

## Discussion

Recognition of aging-related concerns among PLWH, including physical function and sarcopenia, is key to distinguishing individuals at greatest risk for downstream musculoskeletal complications [10]. Our study is the first to explore the relationship between body composition alterations, sarcopenia and physical function among women aging with and without HIV in Latin America and the Caribbean using the rigorous DXA and SPPB assessment tools. We found that women with HIV had significantly lower SPPB scores compared to uninfected women, and also identified key associations between clinical and body composition factors and our outcomes of interest.

**Table 5. Unadjusted and adjusted logistic regression for sarcopenia spectrum, women with HIV (N = 104).**

| Independent variables | Sarcopenia Spectrum[a] | | | |
|---|---|---|---|---|
| | Unadjusted | | Adjusted | |
| | OR | 95% C.I. | OR | 95% C.I. |
| Age | 1.000 | 0.947 to 1.055 | 0.991 | 0.919 to 1.069 |
| Married/cohabited marital status | **0.140** | **0.031 to 0.636‡** | **0.101** | **0.015 to 0.688\*** |
| Education level higher than high school | 1.357 | 0.526 to 3.501 | - | - |
| BMI | **0.572** | **0.447 to 0.732¥** | **0.566** | **0.432 to 0.741¥** |
| Smoke, ever | 0.714 | 0.254 to 2.007 | - | - |
| Current alcohol use | 0.872 | 0.307 to 2.479 | - | - |
| Postmenopausal status | 0.810 | 0.293 to 2.34 | - | - |
| Bone nutritional supplements, ever | 0.702 | 0.075 to 6.571 | - | - |
| Years since HIV diagnosis | 1.008 | 0.939 to 1.083 | - | - |
| Years since ART start | 0.955 | 0.877 to 1.040 | - | - |
| Nadir CD4+ count | 0.998 | 0.995 to 1.001 | - | - |
| Current CD4+ count | **0.998** | **0.996 to 1.000\*** | - | - |
| Viral suppression status | 0.406 | 0.150 to 1.102 | - | - |
| AIDS diagnosis | **10.714** | **2.013 to 57.022‡** | - | - |
| PI exposure | **3.530** | **1.405 to 8.867‡** | **0.191** | **0.050 to 0.728‡** |
| NNRTI exposure | 0.582 | 0.233 to 1.450 | - | - |

[a]*Sarcopenia spectrum: presarcopenia, sarcopenia, severe sarcopenia.*

BMI, body mass index; ART, antiretroviral treatment; PI, protease inhibitor; NNRTI, non-nucleoside reverse transcriptase inhibitor

\*$p \leq 0.05$

‡$p \leq 0.01$

¥$p \leq 0.001$.

HIV epidemiology in Peru has changed significantly over the past decade. From 2014 to 2018, the rate of viral suppression among PLWH receiving ART rose from 36% to 65%, due to widespread free antiretroviral access through health clinics around Peru [24,25]. Consistent with this trend, 78.8% of women with HIV in our study had an undetectable viral load. While improved viral control leads to improved survival among PLWH, data regarding this epidemiological transition among women or men aging with HIV in Peru is completely lacking, and is quite limited from other countries in Latin America and the Caribbean [26,27].

Changes in body composition, such as fat redistribution, have been shown to impact physical function and risk for mortality [1,16]. Since 2013, the International Society for Clinical Densitometry has included ART use in PLWH as one of the indications for using DXA-based body measurements of body composition, in particular for patients receiving agents associated with risk of lipoatrophy [28,29]. Among these changes, peripheral (face and limbs) fat atrophy is commonly reported [30,31]. In our study, women with HIV had lower FM in the limbs and trunk compared to uninfected women, a finding that was statistically significant. Prevalence of fat redistribution varies widely across reports (7–65%) in low- and middle-income countries (LMICs) [32], and in Latin America and the Caribbean, only one study from Brazil has assessed body composition changes in PLWH with objective measurements such as DXA scan. Among 262 PLWH (43% women) the authors found a 40.8% overall prevalence of lipodystrophy based upon DXA-derived fat mass ratio, underscoring the importance of early screening and intervention in this population to prevent complications [33]. Interestingly, in the Brazilian sample, prevalence of lipodystrophy measured by DXA among women was only 15.8%, compared to 48.8% among men (p<0.05), however the authors attributed this finding to their

use of a cutoff for lipodystrophy that was previously standardized in men, but not women, as the prevalence of lipodystrophy among women measured by clinical exam was 56.5% in women, versus 47.7% in men.

In terms of impact of these anthropometric changes on physical performance of PLWH, studies conducted in both upper-income [34–36] and lower-income [37] countries have reported that HIV-infected individuals appear to have worse physical function compared to those without HIV. Among participants in our study, women with HIV had a modestly but significantly lower SPPB score compared to the control group. Compared to another study of 176 PLWH (19% women) in the United States (US) with mean age of 54.6 years, both HIV-infected and uninfected women of our sample had lower SPPB scores [12]. Additionally, 16.4% of HIV-infected and 4% of uninfected women in our study had a SPPB score ≤8 points, which has been shown to be predictive for subsequent disability and mortality [16,38]. A similar study among 65 middle-aged and older (45–55 years) women with HIV in the US with well-controlled disease (mean current $CD4^+$ T-cell 675 cells/mm3, 89% viral load <75 copies/mL) found that 20% of women met criteria for low physical function based upon the SPPB [39]. Even though in our study population, HIV-related biomarkers were relatively stable (mean $CD4^+$ 593.3 cell/mm$^3$; 78.4% viral suppression), a considerable difference was detected in terms of proportion of women with SPPB score ≤8 in the HIV-infected versus uninfected groups. Accordingly, one of the largest studies to apply the SPPB in PLWH found that having well-controlled $CD4^+$ T-cell counts and viral load moderated, but did not eliminate, the odds of reduced physical performance compared the general population [10].

In our study, older age was negatively associated with SPPB score in both women with and without HIV. These findings are consistent with the scarce literature among PLWH conducted mostly in upper-income countries, and to a lesser extent, Africa [12,39,40]. Furthermore, trunk FM was negatively associated with SPPB in the multivariable analyses for both HIV-infected and uninfected women. These results are similar to those reported in previous studies in different aging populations [41–44], however, none of them included HIV-infected individuals. Our findings are consistent with prior evidence suggesting that higher muscle fat infiltration can be an important predictor of physical performance and this relationship has been shown to be stronger in women than in men [45].

In terms of physical strength, upper limb LM was positively associated with grip strength in regression models for both HIV-infected and uninfected individuals, consistent with previous evidence describing muscle mass as an important predictor of physical function [44]. Furthermore, a systematic review and meta-analysis of 30 studies comparing muscle strength and aerobic capacity in people living with HIV compared to matched healthy controls found moderate quality evidence that the lower extremity muscle strength was lower in people with HIV compared to healthy counterparts, and aerobic capacity was comparatively impaired [46]. In our study, although total body LM was slightly higher in uninfected women, differences were not significant. Moreover, among the HIV-infected multivariate model, total body fat percentage was negatively correlated with grip strength, which is also in line with the negative relationship observed in our study between trunk FM and SPPB scores. The mechanisms and natural history of altered lean mass among aging PLWH receiving ART remain poorly understood. Grant et al. described a "return to health effect" where PLWH gained more LM during the first 96 weeks of ART than HIV-uninfected individuals as their immune function and overall health status improves with treatment. However, after that period, patients with HIV had a decrease in LM despite virologic suppression [1]. Other studies have demonstrated a correlation between elevated proinflammatory cytokine levels (e.g. interleukin-6 and C-reactive protein) with low lean mass [47].

Lean or muscle mass is another body compartment frequently affected by HIV [1]. To date, studies on sarcopenia among PLWH are limited and have applied different operational definitions [8]. To our knowledge, ours is the first study that reports sarcopenia among Peruvian women with HIV using the EWGSOP recommendations. The rationale behind this multicomponent definition, which takes into account muscle mass, function and strength, is that characterizing sarcopenia simply in terms of muscle mass significantly limits the clinical utility of the definition [48]. In our study, the majority of women that presented with this syndrome were classified as "sarcopenia" stage. Likewise, a recent metanalysis across 13 studies of PLWH (range: 35–60 years of age) from different geographic regions, including two studies from Brazil, found a pooled sarcopenia prevalence of 13.2% (CI = 5.2–22.9%) using two operational definitions. These included (1) the presence of low LM (four studies) measured by DXA or bioimpedance analysis, or (2) the EWGSOP definition (nine studies) [49]. Our multivariable regression model found that BMI, protease inhibitor (PI) exposure and being married/cohabitant were negatively associated with a sarcopenia spectrum diagnosis. Some of these associations are consistent with previous reports, where higher BMI was associated with a lower odds of sarcopenia [49]. However, although PI exposure was positively associated with sarcopenia in the unadjusted analysis, in the multivariable model it was a protective factor, which contradicts previous data demonstrating increased risk of body composition and metabolic alterations [50,51]. Moreover, in our sample of patients, the proportion of "severe sarcopenia" was higher in the women with HIV compared to uninfected group (3.8% vs 0.9%), however this difference did not achieve statistical significance. This finding warrants further evaluation in larger-scale, longitudinal studies as it may suggest worse prognosis for PLWH if confirmed.

Notably, the EWGSOP published updated guidelines in 2019 [52] that propose a new algorithm for screening and diagnosing sarcopenia, which incorporates the SARC-F screening tool and a slightly modified sequence of the sarcopenia components used in the original guidelines (LM, muscle strength and physical performance). As this new algorithm utilizes European region-specific cutoff thresholds which have not yet been adapted to Latin American populations (whereas in certain regions such as Asia, corresponding guidelines have already been published with thresholds specifically to Asian populations [53]), we opted to apply the well-established 2010 EWGSOP guidelines which have been utilized previously in Latin America and Caribbean populations [54]. Future studies are needed to test and develop appropriate threshold for the new EWGSOP algorithm among individuals from this region.

This study has some important limitations. First, as this was a single-center study our findings might not be generalizable to all women with HIV in Peru, nor to those outside Peru due to differences in cultural and environmental factors. However, this study was conducted in one of the largest public HIV clinics in Peru's capital city. Second, the cross-sectional nature of our study means that associations between physical strength, performance measures, and sarcopenia spectrum outcomes cannot be inferred to suggest causality. Third, the size of our sample was modest, limiting our ability to look at subgroup differences within our outcomes, however our overarching goal was to much needed initial data on body composition, physical function and sarcopenia among women with and without HIV to further promote future studies in this topic among Latin American populations. We also acknowledge the potential for finding associations by chance in the setting of multiple tests done, and therefore took steps in our analytic approach to minimize this risk. Finally, physical activity included in our analyses, and we would recommend future studies to incorporate this important clinical factor.

In summary, as HIV increasingly shifts to a chronic condition, and the proportion of patients aging with HIV globally continues to grow, research characterizing the effect of aging-related comorbidities among this population using well-validated tools is critically needed [55]. Much remains unknown regarding the extent to which alterations in body composition

impact physical function and sarcopenia among women with HIV. We found that women with HIV in our study had significantly lower SPPB scores compared to uninfected women, and that trunk FM and upper limb LM were independent predictors for physical function. These findings underscore the necessity for larger prospective studies among PLWH, particularly in Latin America and the Caribbean, to help identify individuals at greatest risk for declines in physical function and sarcopenia. Our data suggest that targeting lower trunk/abdominal fat mass and increasing limb lean mass could potentially have a positive impact on physical function outcomes. However, intervention studies are needed to inform clinical practice guidelines with regards to optimizing musculoskeletal health among women with HIV.

## Acknowledgments

We thank Debbie Miyasato, Eduardo Matos, Ana Maria Guerrero, Maricruz Huaman, Karim Sanchez, Rocio Galvez and Miriam Santos from the HIV clinic and Radiology Department at Hospital Nacional Arzobispo Loayza for their invaluable help during data collection. Additionally, we greatly thank Cesar Cárcamo for his important input regarding data analysis.

## Author Contributions

**Conceptualization:** Diego M. Cabrera, Mijahil P. Cornejo, Yvett Pinedo, Patricia J. Garcia, Evelyn Hsieh.

**Data curation:** Diego M. Cabrera, Mijahil P. Cornejo, Yvett Pinedo, Patricia J. Garcia, Evelyn Hsieh.

**Formal analysis:** Diego M. Cabrera, Patricia J. Garcia, Evelyn Hsieh.

**Funding acquisition:** Diego M. Cabrera, Patricia J. Garcia, Evelyn Hsieh.

**Investigation:** Diego M. Cabrera, Mijahil P. Cornejo, Yvett Pinedo, Patricia J. Garcia, Evelyn Hsieh.

**Methodology:** Diego M. Cabrera, Mijahil P. Cornejo, Patricia J. Garcia, Evelyn Hsieh.

**Project administration:** Yvett Pinedo.

**Resources:** Diego M. Cabrera, Mijahil P. Cornejo, Yvett Pinedo, Patricia J. Garcia, Evelyn Hsieh.

**Software:** Diego M. Cabrera, Patricia J. Garcia, Evelyn Hsieh.

**Supervision:** Evelyn Hsieh.

**Validation:** Diego M. Cabrera.

**Visualization:** Diego M. Cabrera.

**Writing – original draft:** Diego M. Cabrera, Evelyn Hsieh.

**Writing – review & editing:** Diego M. Cabrera, Mijahil P. Cornejo, Yvett Pinedo, Patricia J. Garcia, Evelyn Hsieh.

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
