## [Decision Letter · Decision Letter 0]

10 Nov 2022

PGPH-D-22-01067

Assessment of Regional Body Composition, Physical Function and Sarcopenia among Peruvian women aging with HIV

Dear Dr. Hsieh,

Thank you for submitting your manuscript to PLOS Global Public Health, and for your patience in us getting back to you. After careful consideration, we feel that it has merit but does not fully meet PLOS Global Public Health’s publication criteria as it currently stands. Therefore, we invite you to submit a revised version of the manuscript that addresses the points raised during the review process.

This is an interesting paper on an important topic from an underreported area; therefore we applaud your work. However, the reviewers have raised some important concerns, which I share. The most important being the overstatement of a difference where there is no statistical evidence for this. Please could you make sure to address this, and adjust your conclusions and interpretations accordingly. We look forward to receiving a new version.

We look forward to receiving your revised manuscript.

Kind regards,

Sabine Hermans

Academic Editor

Journal Requirements:

1. Please send a completed 'Competing Interests' statement, including any COIs declared by your co-authors. If you have no competing interests to declare, please state "The authors have declared that no competing interests exist". Otherwise please declare all competing interests beginning with the statement "I have read the journal's policy and the authors of this manuscript have the following competing interests:"

a. State what role the funders took in the study. If the funders had no role in your study, please state: “The funders had no role in study design, data collection and analysis, decision to publish, or preparation of the manuscript.”

b. If any authors received a salary from any of your funders, please state which authors and which funders.

3. Please indicate by return email the full and correct funding information for your study and confirm the order in which funding contributions should appear.

Additional Editor Comments (if provided):

Reviewers' comments:

Reviewer's Responses to Questions

**Comments to the Author**

1. Does this manuscript meet PLOS Global Public Health’s publication criteria? Is the manuscript technically sound, and do the data support the conclusions? The manuscript must describe methodologically and ethically rigorous research with conclusions that are appropriately drawn based on the data presented.

Reviewer #1: Yes

Reviewer #2: Partly

2. Has the statistical analysis been performed appropriately and rigorously?

Reviewer #1: I don't know

Reviewer #2: Yes

3. Have the authors made all data underlying the findings in their manuscript fully available (please refer to the Data Availability Statement at the start of the manuscript PDF file)?

Reviewer #1: Yes

Reviewer #2: Yes

4. Is the manuscript presented in an intelligible fashion and written in standard English?

Reviewer #1: Yes

Reviewer #2: Yes

5. Review Comments to the Author

Reviewer #1: I think it is crucial to specify in the introduction that HIV infection became a chronic condition because of the antiretroviral treatment.

In the methodology, the authors have to indicate if the status of being on antiretroviral treatment is an inclusion criteria and was is the minimum duration on it for the inclusion in the study. It could be a good point to specify if they have tried to match women of the two group regarding the age groups.

Finally, I deplore the fact that no women was on anti-integrases since it is a recent class of drugs although the cross-sectional character of the study would not have allowed the establishment of a causal link.

Reviewer #2: Review of article PGHPH D-22-01067

This paper presents an original study that aims to analyze the relationship between HIV and body composition changes, physical function and sarcopenia among women from Latin America and Caribbean. This is an interesting study which provides important results on the health impact of chronic HIV. The strengths of this study are: (1) it includes a control group and (2) it uses Dual X-ray absorptiometry (DXA) to assess body composition. On the other hand, the sample size was small with limited power for the analysis and the evaluation of physical function was limited to the SPPB test. Some aspects of the analysis need also to be clarified before the paper could be published. My main comments are listed below.

ABSTRACT

Major comment

The authors should be more careful on their statement about a greater prevalence of severe sarcopenia among HIV+ as it is not significant (only 4 and 2 cases with a p value of 0.8).

Minor comment

The authors may include the study design in the paper’s title.

INTRODUCTION

The authors state that the “musculoskeletal system is significantly affected by HIV and its treatment”, and later explain that this is related to the “decreased bone mineral density, bone mass and [to] heterogeneous disorder of body composition” occurring in HIV-infected people. This section may benefit of being more developed. For instance, why HIV and its treatment result in these negative outcomes and how are they related,

Line 70. What do the authors mean by “the heterogeneity of PLWH”. Do they refer to the heterogeneity of the situations?

It is not clear if body composition changes include sarcopenia as suggested line 67 or not (as suggested line 81).

The objectives presented lines 83 - 84 would benefit to be more specific:

Which relation will be analysed? What are the assumptions to be verified or tested? Or is it a fully exploratory study (which seems not the be the case from the background section).

METHODS

The presentation of the study design, setting and population as well as of the variables used and the measurement methods is clear and detailed enough. On the other hand, the statistical section could be improved.

First, the authors may explain how the study sample size was arrived at. Moreover, the authors used linear regression but did not explain if the normality of the variables was checked and how non-normal distribution was managed. They may also explain if there were missing values and how they were managed.

RESULTS

The paper may benefit of more information on the study flow. For instance, how many participants refused to participate (and what were their characteristics).

Table 2: the difference in the proportion of HIV+ and HIV- participants with SPPB ≤8 should be tested only once overall (and not 2 p-values). P-value for the Chi2 test cannot be 0.09 or <0.0001

Test between the time: if the distribution is not normal (which is likely), a non-parametric test should be used to compare the groups.

Separated analyses were performed for HIV-infected and uninfected participants and by outcomes. Yet, the two groups have different characteristics, thus the results may not be comparable unless some form of standardization is used.

Analyses were also performed separately by outcomes. Was sarcopenia associated with SPPB and/or grip strength?

DISCUSSION

To my view, the discussion needs additional work.

First, it seems to me that there is no significant difference in the prevalence of sarcopenia between the HIV-infected and uninfected groups (although the limited sample size is a problem). Therefore, the authors should be careful in their statement about greater prevalence of severe sarcopenia among HIV-infected women as it is not significant (only 4 and 2 cases, p = 0.8!). The same statement is repeated at the end of the discussion and in the abstract.

I am not sure it is the first study to explore the relationship between body composition, sarcopenia and physical function as it is claimed line 254.

The authors found that women living with HIV have low physical performance, lower bone mass and fat mass, which is an important result that need to be further discussed. For instance, the authors compare their results with two studies conducted in the US (lines 288 to 303). Is this that there are no other studies with SPPB results among women? Otherwise, they should be more systematic. Studies conducted in other LMIC (especially LAC) would be more relevant (e.g. review by Bernard et al 2017 or Gomes-Neto et al, 2018).

I am also surprised to read that there is a scarce literature on the relation between SPPB and age (line 305) as SPPB was initially created to assess physical performance among older people.

Other studies have found that low bone density / OP to be more prevalent that sarcopenia among people living with HIV (e.g., Gregson et al. 2022). The authors may discuss further their results regarding bone density and compare with other studies.

The authors could give possible interpretation of the association between fat trunk and SPPB.

Physical activity is an important factor related to physical performance and body composition that could modify the association found. As it was not assessed, the authors should acknowledge and discuss this limitation. The authors should also acknowledge the multiple tests used and caution that the risk alpha is increased (in other words, some significant associations may just be significant by chance).

Minor comments:

Line 291: do the authors mean “uninfected”?

Line 266: the authors wrote that “Changes in body composition, such as fat redistribution, have been shown to impact physical function and risk for mortality [reference 1].” However, I am not sure that reference 1 addresses physical function? Additional references to support the statement need to be included.

6. PLOS authors have the option to publish the peer review history of their article (what does this mean?). If published, this will include your full peer review and any attached files.

**Do you want your identity to be public for this peer review?** For information about this choice, including consent withdrawal, please see our Privacy Policy.

Reviewer #1: No

Reviewer #2: No

---

## [Editor Report · Decision Letter 1]

13 Jun 2023

Assessment of Regional Body Composition, Physical Function and Sarcopenia among Peruvian women aging with HIV: A Cross-Sectional Study

PGPH-D-22-01067R1

Dear Dr. Hsieh,

We thank you for your patience with us, and are pleased to inform you that your manuscript 'Assessment of Regional Body Composition, Physical Function and Sarcopenia among Peruvian women aging with HIV: A Cross-Sectional Study' has been provisionally accepted for publication in PLOS Global Public Health.

Best regards,

Sabine Hermans

Academic Editor